# Formation of Hydroxyapatite-Based Hybrid Materials in the Presence of Platelet-Poor Plasma Additive

**DOI:** 10.3390/biomimetics8030297

**Published:** 2023-07-09

**Authors:** Ilya E. Glazov, Valentina K. Krut’ko, Tatiana V. Safronova, Nikita A. Sazhnev, Natalia R. Kil’deeva, Roman A. Vlasov, Olga N. Musskaya, Anatoly I. Kulak

**Affiliations:** 1Institute of General and Inorganic Chemistry, National Academy of Sciences of Belarus, Surganova Str., 9/1, 220012 Minsk, Belarus; tsuber@igic.bas-net.by (V.K.K.); musskaja@igic.bas-net.by (O.N.M.); kulak@igic.bas-net.by (A.I.K.); 2Department of Chemistry, Lomonosov Moscow State University, Building, 3, Leninskie Gory, 1, 119991 Moscow, Russia; safronovatv@my.msu.ru; 3Department of Materials Science, Lomonosov Moscow State University, Building, 73, Leninskie Gory, 1, 119991 Moscow, Russia; 4Department of Chemistry and Technology of Polymer Materials and Nanocomposites, Kosygin Russian State University, Malaya Kaluzhskaya, 1, 119071 Moscow, Russia; nsazhnev@mail.ru (N.A.S.); kildeeva@mail.ru (N.R.K.); 5Medical Center “Lode”, Gikalo Str., 1, 220005 Minsk, Belarus; rvalekseevich@mail.ru

**Keywords:** hybrid material, hydroxyapatite formation, amorphous calcium phosphate, platelet-poor plasma, solubility, SBF

## Abstract

Biomaterials based on hydroxyapatite with controllable composition and properties are promising in the field of regenerative bone replacement. One approach to regulate the phase composition of the materials is the introduction of biopolymer-based additives into the synthesis process. The purpose of present study was to investigate the formation of hydroxyapatite-based hybrid materials in the presence of 6–24% platelet-poor plasma (PPP) additive, at a [Ca^2+^]/[PO_4_^3−^] ratio of 1.67, pH 11, and varying maturing time from 4 to 9 days. The mineral component of the materials comprised 53% hydroxyapatite/47% amorphous calcium phosphate after 4 days of maturation and 100% hydroxyapatite after 9 days of maturation. Varying the PPP content between 6% and 24% brought about the formation of materials with rather defined contents of amorphous calcium phosphate and biopolymer component and the desired morphology, ranging from typical apatitic conglomerates to hybrid apatite-biopolymer fibers. The co-precipitated hybrid materials based on hydroxyapatite, amorphous calcium phosphate, and PPP additive exhibited increased solubility in SBF solution, which defines their applicability for repairing rhinoplastic defects.

## 1. Introduction

In modern medicine, repairing damaged and degraded bone tissue remains a problem. The low rate of bone regeneration and the inability of bone itself to repair so-called “critical” defects [1,2] underline the ever-increasing necessity for the development of novel bioactive bone implants. Resorbable materials seem promising due to their complete replacement with newly formed bone tissue soon after implantation [3,4,5]. The resorption of such materials is based on the action of osteoclastic cells, which are capable of creating highly acidic and enzyme-enriched areas on the material surface, causing its fast and complete dissolution [6]. The bioactivity of resorbable materials strongly depends on the balance between their rate of resorption and the rate of new bone formation, and one approach to achieve the balance is to imitate the structure and/or composition of bone.

The inorganic component of bone tissue, bioapatite, is very similar to stoichiometric hydroxyapatite (HAp) Ca_10_(PO_4_)_6_(OH)_2_ in composition and structure [7,8]. However, the resorbability of stoichiometric HAp does not correspond to that of bioapatite, because the latter comprises additional phases, such as amorphous calcium phosphates (ACP) Ca_9_(PO_4_)_6_·nH_2_O, 3.0 ≤ n ≤ 4.5, a metastable precursor of calcium phosphate wet formation [9,10]. Correspondingly, one biomimetic approach to design biomaterials involves the creation of multiphasic calcium phosphates, e.g., via thermal decomposition of calcium-deficient HAp Ca_10−x_H_x_(PO_4_)_6_(OH)_2−x_, 0 ≤ x ≤ 1 [11,12].

Biopolymers perform an important role in the formation of bioapatite, so in situ fabrication of HAp/biopolymer composites is a promising biomimetic approach for obtaining resorbable bone implants with excellent bioactivity. The most interesting biopolymer constituents in such composites include biopolymers of blood plasma, such as albumin, α/β/γ-globuline, fibrinogen, etc. [13]. Blood biopolymers are able to bind and transport proteins, i.e., key growth factors stimulating tissue healing and remodeling [14]. Another attractive feature of blood biopolymers is their availability, which allows them to be used in their most bioactive autologous form. Conventionally, extracted blood plasma undergoes stabilization by a citrate-containing anticoagulant, which prevents blood coagulation, i.e., Ca^2+^-mediated formation, of a dense fibrin clot [14]. Being of autologous nature, citrate-stabilized platelet-poor plasma (PPP) retains its bioactivity and immune safety toward the donor organism and can subsequently be utilized for blood volume resuscitation or tissue regeneration [15,16].

Many blood biopolymers are present in bone tissue, e.g., osteoblastic cells express the blood protein albumin and may adsorb on the surface of apatitic particles, inhibiting the nucleation and growth of HAp crystallites [17]. The ability of blood biopolymers to inhibit apatite crystallization is essential in preventing the calcification of blood vessels [18]. However, this inhibitory activity does not interrupt the normal formation of bioapatite due to the fact that it also abolishes the influence of alkaline phosphatase, an enzyme directly involved in osteoblast function [19].

According to the authors of [20], blood serum additive (i.e., blood plasma without clotting factors) may inhibit the growth of HAp crystals, whereby the predominant inhibitory activity is attributed to a high-molecular fraction of blood serum, i.e., blood biopolymers. However, there are few data on the effect of blood components on the phase composition of precipitated calcium phosphates, even though phase composition is among the properties that strongly affect the material resorbability. According to our previous study [21], the addition of 6% PPP in the precipitation medium brings about the formation of bioactive calcium phosphate composites instead of single-phased stoichiometric HAp. Developing this idea, we decided to expand the range of utilized PPP concentrations and to vary the duration of HAp maturation under the mother solution. In its most dispersed gelatinous form, hybrid materials possess great bioactivity [22], and therefore, the gels comprising HAp and PPP may act as a mineral component of resorbable bone implants, e.g., in combination with the adhesive autofibrin sealant for the treatment of rhinoplastic bone defects [23]. Correspondingly, the purpose of present study was to elucidate the regularities of the wet precipitation of HAp in the presence of 6–24% PPP additive, at pH 11 and maturing time of 4–9 days, with the goal of developing hybrid materials with controllable resorpability which are applicable in rhinoplasty.

## 2. Materials and Methods

### 2.1. Reagents, Additives, and Sample Preparation

The reagents utilized for calcium phosphate preparation included soluble salts of (NH_4_)_2_HPO_4_ (Carl Roth, Karlsruhe, Germany) and CaCl_2_·2H_2_O (Sigma Aldrich, St. Louis, MO, USA), both of which were of analytical grade. The pH values of the precipitation media were regulated by 25% water ammonia (ultrahigh purity, Baza #1 Himreaktivov, Moscow, Russia).

Blood components were extracted from donor blood (20–40-year-old males, B^+^ and B^−^ types), collected with the consent of the donor in accordance with art. 44 of the Health Law of the Republic of Belarus. According to the technique of PPP isolation, 18 mL of donor blood was mixed with 2 mL of 3.8% sodium citrate solution (BelVitunifarm, Vitebsk, Belarus) at a 9:1 volume ratio, followed by centrifugation (3000 rpm, 15 min) and collection of the upper fraction.

The HAp-based samples were prepared in conformity with the existing technique for the precipitation of stoichiometric HAp [12], i.e., by dropwise addition of solution I (0.30–0.34 mol/l HPO_4_^2−^) to solution II (0.40–0.52 mol/L Ca^2+^, 0.24–1.40 mol/L OH^−^, PPP) at pH 11, [Ca^2+^]/[HPO_4_^2−^] at a ratio of 1.67 and at room temperature. In the HAp-PPP samples, the amount of introduced PPP corresponded to 6–24% of the volume fraction of the precipitation medium. The obtained calcium phosphate colloids were matured for up to 9 days, then separated from the mother solution by filtration, repeatedly washed with distilled water, and decanted until a supernatant pH of 7.0–7.5 was achieved.

In order to investigate the dynamics of HAp maturation, we first withdrew a 4 mL aliqot of the HAp colloid after maturation in the mother solution for a period of up to 7 days. We then centrifuged and separated it from the supernatant, washed it with distilled water, and dehydrated it with ethanol and dried it at 400 °C.

### 2.2. Characterization

In order to determine the Ca^2+^ concentration in the supernatant, we utilized complexonometric titrimetry using ethylenediaminetetraacetic acid as a titrant, eriochrome black T as an indicator, 5 mmol/L MgSO_4_ solution as a standard, and NH_4_OH/NH_4_Cl as a buffer solution to keep the pH at 9.

For characterization purposes, the washed colloids were dried at 60 °C until a constant mass of xerogels was achieved, which was then powdered and calcined at 800 °C for 5 h.

The phase composition of the samples was determined by X-ray diffraction (XRD) using an Advance D8 diffractometer (Bruker, Billerica, USA), equipped with Cu_Kα_ radiation (λ = 0.15406 nm), at 20° ≤ 2θ ≤ 45°. The XRD profiles were compared to standards compiled by the Crystallography Open Database (COD), which involved cards for α-polymorph #04-010-4348 and β-polymorph #04-008-8714 of crystalline tricalcium phosphate (TCP) Ca_3_(PO_4_)_2_ and for hexagonal HAp #01-074-0565, using Profex software [24].

To evaluate the content of ACP in the samples, we used an extended variant of the existing technique for the determination of the Ca/P ratio in calcium phosphates by XRD analysis [25]. The original technique relies on thermally induced crystallization of Hap, followed by analyzing the content of crystalline HAp and β-TCP in the formed mixture. In the present study, the key objects of investigation were biphasic calcium phosphate mixtures of HAp and ACP, which, after heat treatment at 800 °C for 5 h, transform into mixtures of crystalline HAp and α-TCP. The following equations describe the calculation of the calcium phosphate composition before heat treatment:χ_ACP_ = (3χ_TCP_)/(1 + 2χ_TCP_), Ca/P = (10 − 7χ_TCP_)/(6 − 4χ_TCP_),(1)
where χ_i_ is mole fraction and Ca/P denotes molar ratio of Ca to P in the solid mixture. The contents of the phases are presented as weight fractions.

The morphological characteristics of the calcium phosphate xerogel surfaces were obtained using a LEO 1420 scanning electron microscope (Carl Zeiss, Oberkochen, Germany), with gold sputtercoating of the surfaces achieved using a K550X sputter coater (Quorum Technologies, Lewes, England).

The functional groups of the powdered samples were studied by Fourier Transform Infrared (FTIR) spectroscopy using a Tensor-27 FTIR-spectrometer (Bruker, Billerica, USA) at an ambient temperature in the range of 1850–450 cm^−1^, by mixing 2 mg of the sample with 800 mg of spectroscopic grade KBr. We used Origin 2018 software to integrate the spectra region at 1850–1360 cm^−1^, representing the integrated area in a.u.

All the XRD patterns and FTIR spectra were normalized. To interpret the XRD and FTIR spectroscopy data of the HAp-based samples, we refer the reader to our previous study [12].

The thermogravimetric analysis was performed on a STA 409 PC LUXX thermal analyzer (NETZSCH, Selb, Germany) in air atmosphere at a heating rate of 10 °C/min and with sample weights of 30–40 mg.

The dissolution curves of the powdered samples were assessed by examining their evolution in a model Simulated Body Fluid (SBF) solution with pH of 7.2 and ion concentrations identical to those in human blood plasma [26]. The SBF solution was prepared in accordance with our previous study [21], using analytical grade reagents (Sigma Aldrich, St. Louis, MO, USA) except for hydrochloric acid, which was prepared from fixanal (vacuum sealed ampoule, AnalysX, Minsk, Belarus). The powdered samples (150 ± 10 mg) were soaked in 50 mL of the SBF solution for 75 days, measuring the supernatant pH value every 7 days. The soaked powders were removed from the supernatant, gently rinsed with distilled water, dried at 60 °C, and weighed.

## 3. Results

In terms of the present study, we considered a HAp to be monophasic and stoichiometric with Ca/P 1.67 if it crystallized at 800 °C without a detectable formation of non-apatitic phases. Certain stoichiometry variations, i.e., resulting from the incorporation of CO_3_^2−^ and HPO_4_^2−^ into apatitic lattice, were not taken into account.

Interpreting the obtained results, we often use the term “maturation”. The term denotes not only the induced presence of the precipitate under the mother solution, but also the corresponding phase transformation which brings about the formation of the most thermodynamically stable phase, e.g., in an alkaline medium, the phase is a stoichiometric hydroxyapatite [12].

According to XRD analysis, heat-treatment at 800 °C induced the transformation of the samples with varying maturing periods, yielding β-TCP (no maturation, Figure 1a, curve 1), 50% HAp/50% β-TCP mixture (30 min, Figure 1a, curve 2), or monophasic HAp (1–7 days, Figure 1a, curves 3–5) in accordance with the following reaction:Ca_10−x_H_x_(PO_4_)_6_(OH)_2−x_ → (1 − x) Ca_10_(PO_4_)_6_(OH)_2_ + 3x β-Ca_3_(PO_4_)_2_,(2)
where 0 ≤ x ≤ 1 denotes calcium-deficiency in the apatite.

At the beginning of maturation, the samples comprised apatitic TCP Ca_9_H(PO_4_)_6_OH, Ca/P 1.50, x = 1, which decomposes at 800 °C into the β-modification of crystalline TCP, and during maturation gradually converts into stoichiometric HAp, Ca/P 1.67, x = 0, that crystallizes at 800 °C without decomposition.

The results of our investigation of the dynamics of calcium phosphate maturing under mother solution (Figure 1b) showed that:

(1) the Ca/P molar ratio of the precipitate increased from the initial value of 1.50 to the equilibrium value of 1.67 over 1 day of maturation;

(2) the concentration of Ca^2+^ ions decreased from 33.4 mmol/l at 30 min of maturing to ~8.6 mmol/l over 4 days.

Both phenomena correspond to the known model of HAp formation [12], including the crystallization of ACP into apatitic TCP and following the reprecipitation of stoichiometric HAp, which is described by the following scheme:Ca_9_(PO_4_)_6_⸱nH_2_O + H_2_O → Ca_9_H(PO_4_)_6_OH + Ca^2+^ + 2OH^−^ → Ca_10_(PO_4_)_6_(OH)_2_,(3)

Basing on the obtained data, we found that a period of 4 days resulted in the complete formation of stoichiometric HAp.

The influence of the PPP additive on the precipitate phase composition was studied on the samples, i.e., those matured for 4 days (with complete formation of individual HAp) and 9 days (extra-maturing for the transformation of PPP-stabilized intermediates). After heating to 800 °C, the XRD pattern of individual HAp, matured for 4 days (Figure 2, curve 1), showed sharp peaks of the crystalline apatite phase; meanwhile, that of HAp/PPP additionally exhibited the formation of up to 45% α-TCP phase (Figure 2, curves 2, 3). The latter is associated with the crystallization of ~47% of ACP at 650 °C, according to reaction (4):Ca_9_(PO_4_)_6_⸱nH_2_O → 3α-Ca_3_(PO_4_)_2_ + nH_2_O,(4)

In contrast, all the XRD patterns of the samples that had matured for 9 days and been heat-treated at 800 °C (Figure 2, curves 4–6) contained only the peaks of the HAp phase. Table 1 summarizes the obtained XRD data.

The observed ACP stabilizing in the HAp/PPP samples, matured for 4 days, presumably, occurred due to the surface adsorption of biopolymer macromolecules on the calcium phosphate particles, which is a common mechanism for the polyelectrolyte-mediated prevention of ACP crystallization [27]. Increasing the PPP concentration from 6% to 24% negligibly affected the amount of stabilized ACP, and the total Ca/P ratio remained at 1.59 (Table 1). One may suppose that the stabilizing effect had already reached its maximum at a PPP concentration of 6%. After continuous maturation for 9 days, the effect of ACP stabilization was no longer observed, implying only a partial inhibition of ACP crystallization into HAp rather than its complete prevention. We associate the latter with the permeability of the adsorbed layer of hydrophilic biopolymers with respect to the water molecules participating in the conversion of ACP into apatite [10].

As shown, under the conditions of HAp synthesis at the initial [Ca^2+^]/[HPO_4_^2−^] ratio of 1.67, the precipitation medium contained Ca^2+^ ions in concentrations of 8–33 mmol/l, which induced PPP coagulation and the formation of fibrin networks throughout the maturation medium. For that reason, the as-formed colloid precipitates of HAp/PPP had reduced flowability compared to those obtained without PPP. During maturation for 4 days, the HAp/PPP precipitates gradually recovered its flowability due to the hydrolytic disruption of fibrin networks at pH 11. For comparison, an individual dense fibrin clot completely dissolves in alkaline solutions (pH 11) for 1 day [21].

According to the SEM images, the xerogels of HAp/6% PPP (Figure 3a,b) were typical for apatite [28], being composed of irregular agglomerates with rough surfaces. For the sample, we observed no evidence of biopolymer structures on the xerogel surface, possibly due to the inclusion of the biopolymer macromolecules into the bulk of the HAp xerogel.

The SEM images of the HAp/24% PPP xerogels (Figure 3c,d) exhibited dense biopolymer fibers up to 30 μm in diameter with rough surfaces, unlike the individual fibrin fibers that have a smooth surface and discernible transparence to an electronic beam [29]. Moreover, the fibers of the HAp/24% PPP samples retained their structural integrity under maturation at pH 11 for 4 days, in contrast to individual fibrin, that completely dissolves in the same conditions. The reason for these differences possibly lies in the fine mixing of the HAp particles and fibrin fibers on the submicron scale. Further proof of this presumption was observed after the mechanical disruption of the fibers, which resulted in the formation of fragments (Figure 3d) with a typical apatitic appearance. Correspondingly, the obtained data implied the formation of a hybrid structure under conditions of simultaneous HAp precipitation and PPP coagulation.

In terms on FTIR spectroscopy, we studied the spectral region at 1850–450 cm^−1^ that represents a set of bands which is sufficient to detect both the HAp and blood biopolymers while also containing no over-broadened bands, i.e., stretching vibrational modes of adsorbed water. Due to the predominantly apatitic nature of the mineral constituent of the samples, all the FTIR spectra after drying at 60 °C (Figure 4a) showed characteristic bands for apatite at 1090, 1040, 956, 603, 566, and 472 cm^−1^ due to O–P–O vibrations and at 633 cm^−1^ due to O–H libation. The PPP-free samples (Figure 4a, curves 1, 4) had bands at 1650 cm^−1^ due to the H–O–H bending vibrations of adsorbed water, as well as bands at 1500–1350 cm^−1^ due to the O–C–O stretching vibrations of CO_3_^2−^ ions in the apatite lattice. The integrated intensity of these bands of 13 ± 5 a.u. mostly arose from vibrations of apatitic CO_3_^2−^ ions that were incorporated in the precipitate structure during maturation in an alkaline solution. For HAp precipitation in a highly alkaline, CO_3_^2−^-poor medium, the incorporation proceeded via substitution of PO_4_^3−^ for CO_3_^2−^ (B-type), for which one of the most intense band is located at ~1480 cm^−1^ [30].

In the case of the PPP-containing samples (Figure 4a, curves 2, 3, 5, 6), the spectral region at 1850–1350 cm^−1^ was mainly composed of bands assigned to vibrations of biopolymer functional groups [31]: amide I—1660, 1640 cm^−1^; amide II—1540 cm^−1^; and hydrophobic side groups of amino acids—1450, 1415 cm^−1^. We did not consider low-intensity amide III bands at 1330–1190 cm^−1^ due to their overlapping with intense apatitic O–P–O bands. One may notice the gap in the spectra at 1480 cm^−1^ (Figure 4a, denoted by a dashed line), where the previously mentioned B-type carbonate band would be expected to overlap with the biopolymer bands. The integrated area of the biopolymer bands at 1850–1350 cm^−1^ was related to total biopolymer content in the samples. As shown in the obtained values, the four-fold increase of the introduced PPP concentration from 6% to 24% provided comparably less increasing of the total biopolymer amount entrapped by solid precipitate, i.e., from 24 ± 3 a.u. to 38 ± 3 a.u. One explanation of this refers to the partial removal of biopolymers during washing–decantation, due to which the HAp/6% PPP sample (apatitic morphology) and the HAp/24% PPP sample (biopolymer-like morphology) comprised comparable total amounts of biopolymers.

In order to verify the reliability of the regularities revealed by FTIR spectroscopy, we performed a thermogravimetric analysis of the samples. According to the data obtained (Figure 4b), the HAp-based samples underwent three-staged thermal transformations:

I. The dehydration stage was observed at 25–250 °C, accompanied by weight loss of 5.4–9.8%, corresponding to the elimination of absorbed water and the decomposition of the HAp crystallohydrate [32].

II. Further heating of the PPP-based samples caused biopolymer decomposition at 250–750 °C. For the individual HAp (Figure 4b, curve 1), the mass decrease of 2.7% was due to the removal of residual water, but in the case of the HAp/PPP samples (Figure 4b, curves 2, 3), the mass loss values correspond to the weight content of biopolymer and organic substances in the samples. The increase of the introduced PPP amount from 6% to 24% resulted in an increase in the content of decomposing substances from 18.6% to 22.5%, which is in accordance with the FTIR data (Figure 4a).

III. Crystallization of calcium phosphates occurred at 750–1000 °C. The corresponding mass loss was maximal (0.6%) for the individual HAp (Figure 4b, curve 1); this was explained by the gradual decomposition of lattice CO_3_^2−^ ions with the removal of CO_2_ [30]. For the HAp/PPP samples (Figure 4b, curves 2, 3), one may observe the abrupt mass loss of 0.2% (pointed by arrows) at 750–780 °C due to the crystallization of ACP, according to reaction (4). Above 780 °C, the mass of the HAp/PPP samples remained virtually unchanged, implying insignificant decomposition of CO_3_^2−^ ions. As shown, the degree of carbonation in the HAp/PPP was much less than that in the individual HAp, which may explain the absence of carbonate bands in the FTIR spectra of the HAp/PPP. We associated the prevention of CO_3_^2−^ incorporation in the apatitic lattice with the barrier effect of the adsorbed biopolymers of PPP.

The dissolution of the hybrid materials in a model SBF solution was studied on the example of the powdered HAp/6% PPP sample, using individual HAp as a reference. According to the data obtained (Figure 5), the pH value of the supernatant increased over 28 days of soaking; this may be attributed to both the dissolution of the HAp and the substitution of apatitic OH^−^ ions with Cl^−^ ions from the SBF, according to the following reactions:Ca_10_(PO_4_)_6_(OH)_2_ → 10Ca^2+^ + 6PO_4_^3−^ + 2OH^−^, (5)
Ca_10_(PO_4_)_6_(OH)_2_ + xCl^−^ → Ca_10_(PO_4_)_6_(OH)_2−x_Cl_x_ + xOH^−^, (6)

After 28 days of the submerging, the pH values only fluctuated over an average of ~7.6. In the case of the HAp/6% PPP sample, the fluctuations reached ∆pH ± 0.10, compared to ∆pH ± 0.03 for the individual HAp. In our previous article [21], we demonstrated the transformation of HAp/ACP mixtures into the highly stoichiometric apatite during soaking in SBF. Therefore, the significant pH fluctuations may have been associated with the transformation of the ACP into HAp accompanied by pH lowering in correspondence to scheme (3).

The overall mass decrease of the sample after the soaking was 9% for the individual HAp and 17% for the HAp/6% PPP. The greater decrease of the mass of the PPP-based material may be attributed to the following factors:

(1) the increased solubility [12] of the metastable ACP (pK_S_ ≈ 25) in relation to the HAp (pK_S_ ≈ 117);

(2) the elaboration of the biopolymer macromolecules under the conditions of calcium phosphate reprecipitation [33].

Therefore, the hybrid HAp/PPP material showed increased solubility under model physiological-like conditions compared to a monophasic stoichiometric HAp.

## 4. Discussion

The current study focuses on the alteration of the phase composition of calcium phosphates formed in the presence of PPP. PPP comprises macromolecules, such as albumin and fibrin, which are able to adsorb on the particle surface of the forming calcium phosphates [18]. In terms of crystal growth kinetics, the influence of adsorbed blood biopolymers, i.e., albumin [17], is known to partially inhibit the HAp crystallization, bringing about the formation of smaller crystallites. This phenomenon is attributed to the limitation of Ca^2+^, PO_4_^3−^, and OH^−^ ion transport to the surface of the growing particles [34,35]. However, the maturation of HAp comprises not only ion transport, but also a calcium phosphate cluster rearrangement, surface reprecipitation, etc., which result in the transformation of the intermediate phases (ACP, apatitic TCP) into the stable ones [33]. It is known [10,27] that some polyelectrolytes may stabilize ACP via surface adsorption, but the same phenomenon was not previously studied for the blood biopolymers.

According to the obtained results, the additive of PPP may partially inhibit the transformation of the ACP into the HAp. In this sense, the biopolymer macromolecules appear to be entrapped in the structure of the calcium phosphates. Both phenomena contribute to increasing the solubility of the hybrid material, due to which it may be utilized as a more soluble analogue of stoichiometric HAp.

While most studies in this field have focused on the physical–chemical properties of the powdered xerogel form of hybrid HAp/PPP materials due to their usability, we considered the bioactive gel form of such materials to be the most suitable for practical application (Figure 6). Calcium phosphate gels have great chemical and biological activity due to their high specific area; however, they lack structural integrity. Nonetheless, in terms of the treatment of defects of non-load-bearing bone [36], the key property of a biomaterial is resorbability, rather than mechanical properties.

One possible route for compensating for the flowability of the HAp-based gels is mixing it with an excess of PPP in the presence of Ca^2+^ ions [23], which results in the formation of an adhesive composite clot (Figure 6). Such composites are intended to fill small bone defects of any shape, and utilizing the hybrid mineral component may enhance the bioactivity of the material. Moreover, fibrin clots can hold autograft fragments of bone or cartilage. Therefore, introducing hybrid HAp/ACP/PPP gel into an adhesive composite would be beneficial for rhinoplasty.

Similar to our previous study [21], an additive of PPP was chosen as a biopolymer source in hybrid gels in order to eliminate the possible influence of platelets on HAp formation. Another possible source of blood biopolymers is a platelet-rich plasma (PRP), which is blood plasma enriched with platelets [37]. PRP is known to greatly enhance tissue regeneration, both individually and in compositions with HAp-based hybrid materials [38]. However, in terms of the present study, the highly alkaline conditions (pH 11) of the HAp-based hybrid gel precipitation may induce a hydrolytic disruption of blood cells [39], thereby abolishing the advantage of PRP over PPP. On the other hand, PRP may substitute PPP at the stage of the adhesive composite formation (Figure 6), increasing the healing potency of the material, while PPP remains applicable for resource-poor settings [40]. A comparison of adhesive composites based on hybrid gel and PPP or PRP is a promising direction for expanding the ideas presented in present study.

## 5. Conclusions

The formation of HAp in the presence of PPP was studied under conditions of co-precipitation at pH 11, a [Ca^2+^]/[PO_4_^3−^] ratio of 1.67, and a PPP volume fraction of 6–24%. The effect of PPP on HAp formation partially inhibited the crystallization of ACP and the incorporation of CO_3_^2−^ into the apatite structure, which increased the formation of ACP composites of HAp by up to 47% after 4 days of maturing. At the same time, increasing the introduced PPP content from 6 to 24% insignificantly affected the amount of stabilized ACP, whereas continuous maturation for 9 days led to the formation of monophasic HAp. Varying the PPP content in the range of 6–24% resulted in the formation of materials with rather predictable biopolymer contents of 18.6–22.5% and desired morphologies, ranging from typical apatitic conglomerates to hybrid apatite-biopolymer fibers. The co-precipitated hybrid materials based on HAp and PPP showed increased solubility in SBF solution, which is an important factor in terms of their applicability for repairing rhinoplastic defects.

## Figures and Tables

**Figure 1 biomimetics-08-00297-f001:**
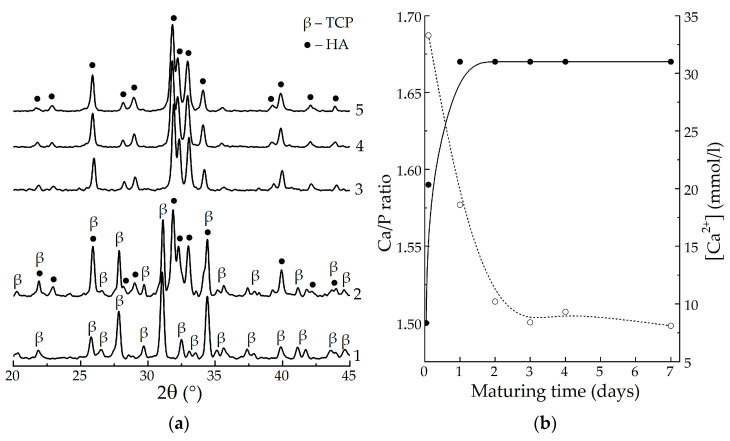
XRD patterns of the samples, matured for different periods, after heat-treating at 800 °C (**a**): 1—no maturation; 2—30 min; 3—1 day; 4—4 days; 5—7 days; changing of the calculated Ca/P molar ratio of the precipitate (solid) and Ca^2+^ concentration in the precipitation medium (dash) during maturation (**b**).

**Figure 2 biomimetics-08-00297-f002:**
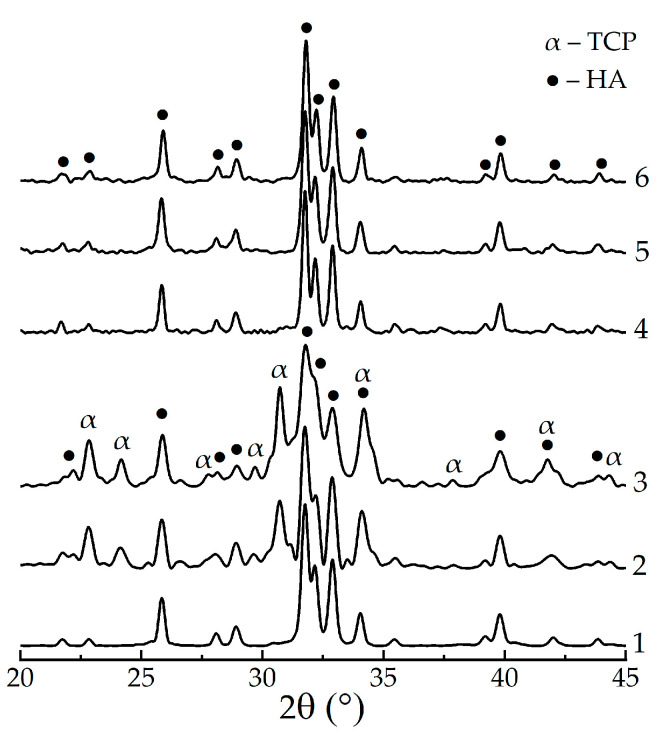
XRD patterns of the powdered xerogels after heat-treating at 800 °C: 1—HAp, 4 days; 2—HAp/6% PPP, 4 days; 3—HAp/24% PPP, 4 days; 4—HAp, 9 days; 5—HAp/6% PPP, 9 days; 6—HAp/24% PPP, 9 days.

**Figure 3 biomimetics-08-00297-f003:**
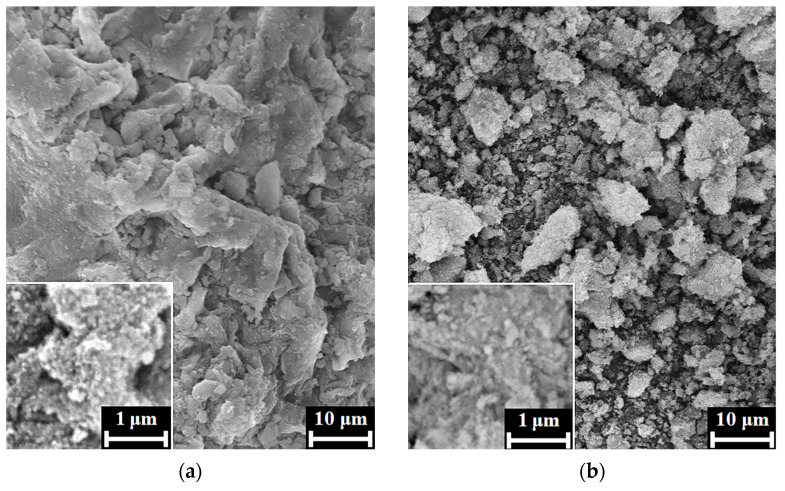
SEM images of the xerogel surfaces of HAp/6% PPP (**a**,**b**) and HAp/24% PPP (**c**,**d**), matured for 4 days, after drying at 60 °C, before (**a**,**c**) and after (**b**,**d**) mechanical disruption.

**Figure 4 biomimetics-08-00297-f004:**
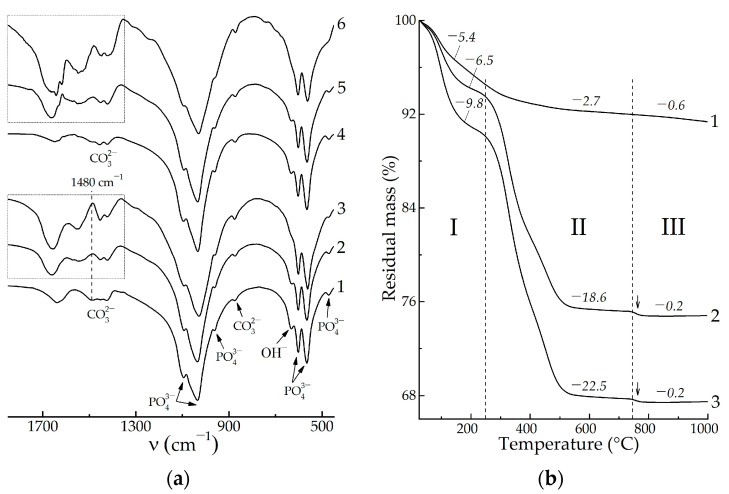
The FTIR spectra (**a**) and the thermogravimetric curves (**b**) of the powdered xerogels after heat-treating at 60 °C: 1—HAp, 4 days; 2—HAp/6% PPP, 4 days; 3—HAp/24% PPP, 4 days; 4—HAp, 9 days; 5—HAp/6% PPP, 9 days; 6—HAp/24% PPP, 9 days. Arrows denote abrupt mass decrease due to ACP crystallization. Latin numerals (I–III) denote different stages of the thermal transformations.

**Figure 5 biomimetics-08-00297-f005:**
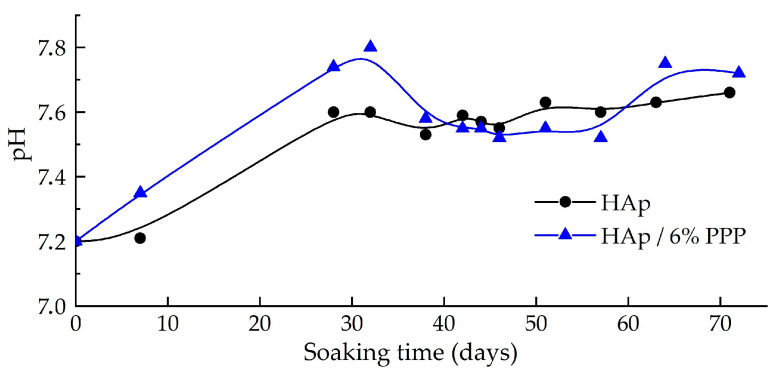
pH dynamics during the soaking of the samples in SBF solution for 75 days.

**Figure 6 biomimetics-08-00297-f006:**
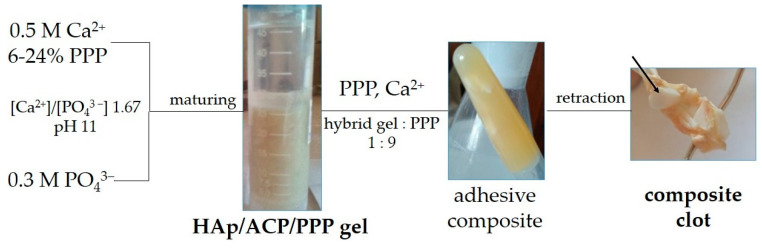
Schematic for the preparation of the hybrid HAp/ACP/PPP gel and its further involvement in obtaining an adhesive composite for maxillofacial surgery. The arrow denotes a fragment of an autograft fixed in an autofibrin clot structure.

**Table 1 biomimetics-08-00297-t001:** Results of the XRD phase analysis of the HAp/PPP samples.

Sample	Maturing Time, Days	Content of Phases, %		Ca/P
		ACP (60 °C) ^1^	α-TCP (800 °C)	
HAp	4	–	–	1.67
HAp/6% PPP		46	44	1.59
HAp/24% PPP		47	45	1.59
HAp	9	–	–	1.67
HAp/6% PPP		–	–	1.67
HAp/24% PPP		–	–	1.67

^1^ Calculated in accordance to Section 2.2; “–“ denotes the absence of the phase in the sample.

## Data Availability

Not applicable.

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
