# Peer review of "Formation of Hydroxyapatite-Based Hybrid Materials in the Presence of Platelet-Poor Plasma Additive"

_biomimetics, 2023, doi:10.3390/biomimetics8030297_

Round 1
Reviewer 1 Report
attached

Moderate editing of English language required
Reviewer 2 Report
This study investigated the fabrication of hydroxyapatite with the presence of platelet-poor plasma and optimized the concentration of PPP and the synthetic condition. The following comments are provided for authors’ consideration:
1. In introduction, could the authors explain more on the necessity of this study? as there have been similar papers published. https://doi.org/10.1016/j.mtcomm.2021.102224
2. SEM-EDS is suggested to visualize the chemical composition and distribution on the surface.
3. It is suggested to add the picture of the final product as the authors proposed to use the product for bone regeneration, is the product in powder form or feasible to be fabricated in various shapes?
4. As per the instructions for authors of this journal, discussion part is required.
5. Some of the references are more than 20 years ago. The authors are suggested to use recent publications as references.
Round 2
Reviewer 1 Report
please see attached document

Moderate editing of English language required
Reviewer 2 Report
No further comment.
Author Response
We would like to thank the reviewer again for providing valuable suggestions to improve the quality of the manuscript